# Protein Supplementation as a Nutritional Strategy to Reduce Gastrointestinal Nematodiasis in Periparturient and Lactating Pelibuey Ewes in a Tropical Environment

**DOI:** 10.3390/pathogens11080941

**Published:** 2022-08-19

**Authors:** Yoel López-Leyva, Roberto González-Garduño, Alvar Alonzo Cruz-Tamayo, Javier Arece-García, Maximino Huerta-Bravo, Rodolfo Ramírez-Valverde, Glafiro Torres-Hernández, M. Eugenia López-Arellano

**Affiliations:** 1Postgraduate in Animal Production, Universidad Autónoma Chapingo, Texcoco 56230, Estado de Mexico, Mexico; 2Southeast Regional University Unit, Universidad Autónoma Chapingo, Teapa 86800, Tabasco, Mexico; 3Faculty of Agricultural Sciences, Universidad Autónoma de Campeche, Escárcega 24350, Campeche, Mexico; 4Pasture and Forage Experimental Station “Indio Hatuey”, Universidad de Matanzas, Matanzas 44280, Cuba; 5Colegio de Postgraduados, Campus Montecillo, Montecillo 56230, Estado de Mexico, Mexico; 6Department of Helmintology, National Center for Disciplinary Research in Animal Health, INIFAP, Jiutepec 62560, Morelos, Mexico

**Keywords:** nutrition, parasitism, peripartum rise, sheep

## Abstract

The study was carried out to evaluate the effect of energy and protein supplementation on parasitological and hematological response during peripartum and lactation of productive and non-productive Pelibuey ewes in a tropical environment. Forty-eight Pelibuey ewes aged 3–5 years and with a body weight of 31 ± 5 kg were used. Four groups of 12 ewes, including non-pregnant and productive ewes, were formed. A factorial treatment design was formulated, where two levels of energy (low, 9.6 MJ/kg, *n* = 24; and high, 10.1 MJ/kg, *n* = 24) and two levels of protein (high, 15% crude protein in diet, *n* = 24; and low, 8% crude protein in diet, *n* = 24) were studied. Fecal and blood samples were collected to determine the fecal egg count (FEC) of gastrointestinal nematodes (GIN), packed cell volume (PCV) and peripheral eosinophil (EOS) count. These variables were rearranged with respect to the lambing date in a retrospective study. The high dietary protein level had a significant effect on reducing the FEC and increasing the PCV of ewes during lactation, in comparison with animals fed with the low protein level. Differences in the study variables were attributed to physiological stage. Lactating ewes showed the highest FEC values (2709 ± 359 EPG), the lowest PCV values (21.9 ± 0.7%) and the lowest EOS (0.59 ± 0.6 Cells × 10^3^ µL). It is concluded that high levels of dietary protein improve the hematological response and reduce the FEC in Pelibuey ewes under grazing conditions. The non-pregnant ewes maintained some resilience and resistance to GIN infection compared to productive ewes.

## 1. Introduction

Gastrointestinal nematodes (GIN) are considered one of the most important diseases in grazing sheep [1]. *Haemonchus contortus* is the most prevalent of all reported GIN species in warm climates [2]. This is a pathogenic parasite that causes reduction in food intake [3] weight loss, severe anemia, ulcerations, congestion of the abomasal mucosa, hemorrhagic foci, hyperplasia of the glands with infiltration of lymphocytes and eosinophils [4], deterioration of body condition and high mortality in susceptible sheep [5,6].

During the productive life of ewes, the periparturient period is very important because they suffer a reduction in immunity associated with the stress of lambing and lactation, which, in turn, causes favorable conditions for the development of a GIN parasitic phase [7], and, as a consequence, high fecundity of GIN females, which results in high fecal egg counts (FEC) and increases the susceptibility to other infections [8]. This phenomenon is well known as peripartum relaxation of immunity (PPRI), peripartum rise (PPR) or spring rise [9]. Due to this situation, ewes during peripartum carry the main responsibility for the contamination of pastures with GIN eggs [8], while non-pregnant ewes maintain a high immune response without showing an increase in FEC and an adequate PCV.

In the PPR, ewes show physiological and metabolic changes [10] that considerably increase their nutrition requirements for milk production for feeding one or more lambs [11], in addition to the requirements to deal with GIN infection. Thus, nutritional management is vital to enable ewes to get through this critical period and ensure defense mechanisms against infectious and parasitic pathogens [12], in addition to producing strong and healthy lambs. It is well documented that a balanced diet reduces the magnitude or duration of the peripartum rise by improving the immune response of the ewes because the PPR has a nutritional basis [3]. Protein metabolism is what is most affected during infection with GIN, due to the reduction of food intake, loss of protein and blood in the digestive system and the need for the repair of affected tissues throughout [13]. Therefore, protein supplementation in the feed can help to improve the resistance and resilience of the host against parasites [14]. Infected animals require more energy to maintain an increase in body weight similar to that of non-infected animals for the development of immunological mechanisms [15]. Due to this situation, the comparison of productive ewes with a high infection with non-productive (not pregnant) ewes that have low fecal counts is important in that the comparison allows the determination of the effect of nutrients in each of the physiological stages. Therefore, the proposed hypothesis assumes that increasing protein and energy concentration in the diet helps to reduce the effects of parasites and thereby reduce the FEC in grazing sheep, mainly during lactation or peripartum, which allows the generation of management recommendations to reduce damage to animal health. The objective of the present study was to evaluate the parasitological and hematological response of adult productive and non-productive Pelibuey ewes in grazing during peripartum and lactation supplemented with two levels of dietary energy and protein.

## 2. Results

### 2.1. Parasitological and Hematological Response during Peripartum

*H. contortus* (70%) was the main GIN observed in the coprocultures, *Trichostrongylus colubriformis* (20%) and *Strongyloides papillosus* (10%) were found. This study showed a long PPR response in the ewes. The FEC increased to values higher than 1800 EPG three weeks before lambing, and a peak was observed at six weeks of lactation (4950 EPG). High FEC were maintained during the five weeks of lactation and no EPG reduction was observed at the end of lactation. Throughout peripartum the FEC was higher in lactating ewes than in the non-pregnant group (Figure 1).

The highest PCV was observed in non-pregnant ewes, and similar values to productive ewes were observed at week 18 of gestation, but later the PCV in productive ewes tended to decrease gradually as it approached parturition and during the following weeks of lactation, with a slight tendency to increase from weeks 6 to 7 of lactation (Figure 2).

In the pregnant ewes, the EOS counts was similar to non-pregnant ewes (Figure 3), but during lactation the EOS counts were drastically reduced.

The difference in the study variables (*p* < 0.01) due to the effect of physiological stage when comparing productive against non-productive ewes was evident in the sixth week of lactation (Table 1). Lactating females had an increase in FEC (4950 EPG) of 1562% compared to non-productive females (298 EPG) and a 44.9% reduction in PCV (non-productive 29.4% vs. lactating 16.2%), while the maximum reduction in EOS count occurred one week earlier with a 78% reduction (non-productive 0.95 vs. lactating 0.21 Cells × 10^3^ µL).

### 2.2. Effect of Energy and Protein

In the present study, ewes receiving the high protein level (15% crude protein in diet) showed lower FEC and higher PCV values (*p* ≤ 0.05) compared to ewes feeding at the low protein level (8% crude protein in diet). As expected, important differences in the study variables were attributed to physiological stage and the PPR was appreciated in the lactating ewes which showed the highest FEC values (2709 ± 359 EPG), the lowest PCV values (21.9 ± 0.7%) and the lower EOS (0.59 ± 0.6 Cells × 10^3^ µL). The non-productive ewes group did not show differences in PCV and EOS in comparison to the pregnant ones. The proposed energy level in the diet had no effect on the EPG, PCV or EOS (Table 2).

### 2.3. Interaction of Energy and Protein in the Physiological Stage

The interaction of dietary protein in the physiological stage was evident in FEC and PCV values. Thus, lactating ewes fed the high level of protein (15% crude protein in diet) showed a reduction in FEC and better PCV than ewes receiving low protein (8% crude protein in diet) (Figure 4). It was noteworthy that the limiting factor was the protein with a metabolizable energy level of 9.6 MJ/Kg in the lactating ewes, since those that received the high level of dietary protein had lower FEC and higher PCV levels (*p* < 0.01) than the lower-protein-level group, while in pregnant and non-productive ewes no effect of dietary protein was observed (Figure 4).

## 3. Discussion

The results obtained in the PCV in this study are related to the presence of *H. contortus*, which is a hematophagous nematode that causes anemia. The confirmation of the high prevalence of this species has been similar to other studies in the same region, where *H. contortus* was the main nematode species identified, with a prevalence over 90%; and *T. colubriformis* were also found [16]. In our study *S. papillosus* was found but not *Oesophagostomum* spp.

The study of energy and protein supplementation in sheep is important because it represents a sustainable alternative to reduce the impact on the health of sheep infected with GIN. Proper nutrition overcomes the low effectiveness of anthelmintics caused by the increasing drug resistant level in *H. contortus* populations. In addition, this species is a high pathogenic hematophagous nematode [17] responsible for a reduction in the PCV, leading to anemia. During lactation, a reduction of immunity occurs, as suggested in our study by the low peripheral eosinophils counts, particularly in productive and heavily infected animals, which in turn favors an increase in the FEC [6]. For this reason, the search for alternatives to minimize the damage of this species is a priority. Supplementation alternatives must also be sustainable with ingredients of the region such as sugar cane, which, processed and enriched, provides enough nutrients to increase the immunity of the animal while maintaining the productive level of milk for raising lambs [18].

Many studies have been carried out to measure the effect of nutritional manipulation in the control of gastrointestinal parasites [19,20,21,22]. The response during peripartum of the variables studied and the comparison of productive ewes (lactating) against non-productive ewes gives a clear indication of the limiting of dietary nutrients, which, in this case, was protein. It was hypothesized that both energy and protein are needed, especially during lactation. However, in the study, it was very clear that when energy levels are adequate, protein is the limiting factor, so the simple effect of protein was clearly shown when the FEC was reduced by increasing the protein level, but only in lactation; although, along with the amount of protein given, some authors suggest that the parasitological response is regulated also by the quality of protein supplemented [22,23].

The effect of dietary protein on the reduction in FEC has been attributed to a replacement of nutrients that are diverted to develop the protective elements of immunity [24]. Protein losses due to the effect of GIN infection can be corrected through the intake of dietary protein, and, for this reason, protein requirements are higher than currently recommended for non-parasitized animals [22]. Méndez-Ortíz et al. (2019) [15] suggest an important relationship between parasitism and dietary energy and protein consumption. Unfortunately, consumption was not evaluated in this study because, in addition to the supplement, the sheep grazed, so the estimation of energy and protein consumption was not carried out. An additional energy supply is required, in addition to protein, for the development of immunological mechanisms [25]. However, in the present study, the energy effect was not observed in any of the variables studied, which could suggest that the level used is sufficient to develop the main physiological functions, perhaps the long rest period (18 h) during housing was the reason why the energy level was not as important in nematode control as other studies have indicated [25]. Apparently, energy supplementation is subject to a process of metabolic reprogramming necessary for the development and functioning of the immune system [26]. This can generate a diversion of energy to the synthesis of the cellular metabolites necessary for defense processes [27,28]. It is possible that the effects of energy may not have been observed because protein and energy interact and dietary protein is also a source of energy. That is why the use of dietary protein is used more efficiently in resilience and resistance against GIN [24].

This study provided evidence, as in other studies, about the increased FEC during the last weeks of pregnancy and the first weeks of lactation [29]. The main explanation is from the breakdown of the immunological system that provokes an increased FEC. This is associated with several factors that act simultaneously in ewes, such as an energy imbalance, an aspect that could not be seen in the present study because the energy level had no significant effects on the three variables studied (FEC, PCV, EOS). Another factor is the reduction in the feed intake combined with a low absorption capacity of the intestine that causes poor nutrient use [14]. For the aforementioned, the peripartum rise is linked to increased energy and protein needs for fetal growth, milk production, and tissue repairs [30,31]. Other important aspects associated with the high FEC are explained by the massive development of hypobiotic larvae [32] during the immunosuppression that occurs at this stage, which favors parasitic development [33].

The PCV is an important indicator of health status in sheep because it is closely related to the presence of anemia [34]. Its decrease is related with the infection of blood-sucking nematodes such as *H. contortus* [35]. The PCV is affected by a high GIN burden [34] and is considered one of the criteria of resilience and genetic resistance of sheep [21], for this reason it is included in genetic selection programs of flocks. Along with a reduction in PCV that occurs during infection, other elements of the immune system are affected, since it has been observed that the number of neutrophils is reduced [36]. In the present study, a breakdown of immunity during lactation was noted, when eosinophils were drastically reduced after the third week of lactation, coinciding with the reduction in PCV. Protein supplementation helps to reduce FEC, specifically in such critical moments as the last third of pregnancy and lactation and, together with energy supplementation, can improve the immune response of ewes in the peripartum [37]. Hematocrit decreases progressively from the beginning of pregnancy, reaching the lowest levels in the second month of lactation [38].

Eosinophils are defense cells and act together with immunoglobulins in the control of GINs [39,40]. Eosinophils, mast cells and globular leukocytes are effector cells that can selectively secrete cytokines, chemokines and other mediators in response to helminth infection, contributing to both effector functions and the maintenance of homeostasis [41]. In the present study, the physiological stage affected EOS counts. In peripartum, an increase in the FEC occurred, which triggers the classic signs of eosinophilia together with increases in macrophages and dendritic cells, which contribute to the defense of the host against the challenge [42]. However, the levels of eosinophils were reduced after the fourth week of lactation, reaching their lowest values at the fifth week after lambing, and from the sixth week normal values of eosinophils were counted. Eosinophilia has commonly been reported to be negatively correlated with FEC, which is reflected in the high counts when a lower parasite burden occurs [43], a situation that was observed in non-pregnant ewes with lower FEC and higher EOS counts. In relation to productive ewes, in pregnant and lactating ewes an increased level of eosinophil accumulation in tissues or blood, with marked degranulation, can be seen in parasitic infections [41]. However, the breakdown of immunity implies a reduction in the EOS count in the middle and at the end of lactation, as observed in the present study.

## 4. Materials and Methods

### 4.1. Location

The study was performed from October to December 2019 at the Experimental Station of Pastures and Forages (EEPF) “Indio Hatuey” of the Matanzas University in Cuba. The climate of the region is Equatorial Savannah with a dry summer [44].

### 4.2. Ewe Management

Forty-eight Pelibuey ewes, 3–5 years old and with a body weight of 31 ± 5 kg were used. Ewes from each group were randomly assigned to four experimental groups (*n* = 12 per group, 6 pregnant and 6 non-pregnant or non-productive) according to a factorial treatment design (high energy–high protein, high energy–low protein, low energy–low protein, and low energy–low protein). Only the weights of the sheep were used to have homogeneity in the groups. Previously, during the mating period, 30 ewes remained with the rams and another 30 ewes were separated without mating (non-productive ewes). After the mating period, all ewes remained in a single group on grazing. At the beginning of the sampling, only 24 females from each group were used, discarding the ewes that were not pregnant or that had some problem. At the beginning of mating, both productive ewes and non-productive ewes were dewormed with Levamisole 10% (Levamisole hydrochloride at 7.5 mg/kg BW, Labiofam, Cuba). The study was retrospective because the gestation week was determined at the lambing date. The lambing date of each ewe was used to categorize the pre-partum period, which started at the 13th week of pregnancy. The lactation period was considered from the lambing day up to 49 days postpartum (subdivided into 7 weeks). From week 13 to 15 of pregnancy, an adaptation period to the experimental diets was provided. The ewes were kept in rotational grazing for 6 h per day at a stocking rate of 20 ewes per hectare. The paddocks had a mixture of *Dichantium annulatum* and *Megathyrsus maximus*. After grazing, the animals were housed and separated individually into pens and supplemented with the experimental dietary treatment and water *ad libitum*. The ewes remained during the afternoon and night in each assigned pen along with their lambs.

### 4.3. Diets and Treatments

The experimental diets included sugar cane to take advantage of the local food resources, in addition other local ingredients were used to increase the energy or protein in diet. The combination of two levels of energy (high and low) and two levels of protein (high and low) represented the factorial design experimental treatments (Table 3). The diets were processed through solid-state fermentation [45]; briefly, the elaboration process consisted of cutting the cane one day before its use, then grinding in a forage grinder (Power 11 HP, model PFA 3000). Immediately, all the ingredients were mixed, and it was spread out on a flat surface and left to ferment for 24 h. At the end of this process, it was bagged for conservation for a period of 8 to 12 days.

The diets were provided in two fractions, one in the morning and one in the afternoon, and, in total, 3 kg were allocated on a fresh basis per sheep to allow a rejection of at least 15%.

Food samples were taken per treatment for dry matter (DM) determination in an oven at 60 °C for 72 h. Then, each sample was stored in plastic bags. Samples were ground and the composition was determined as indicated in AOAC International (2016) [46]; briefly, ash at 600 °C for 2 h to calculate organic matter (OM) then ether extract by the method of acid hydrolysis using petroleum ether as a solvent (EE). In addition, total nitrogen was determined with an Organic Analyzer Flash 2000 (Thermo Scientific, Cambridge, United Kingdom), neutral detergent fiber (NDF) and acid detergent fiber (ADF) by the procedures described by Van Soest [47]. Metabolizable energy was determined according to Galyean et al. [48].

### 4.4. Sampling, Parasitological and Blood Analysis

Weekly fecal samples were collected directly from the rectum to determine the number of eggs per gram of feces (EPG) by the flotation McMaster technique [49] with a minimum sensitivity of 50 EPG. Blood samples were also collected weekly by jugular vein puncture with vacutainer tubes with ethylenediaminetetraacetate (EDTA) as anticoagulant (Vacutainer^TM^; BD Biosciences, Franklin Lakes, NJ, USA). The whole blood was used to determine the packed cell volume (PCV) using the classical microhematocrit technique, and peripheral eosinophils count (EOS) using a Neubauer chamber [50].

A copro-culture from fecal samples was made at the beginning and at the end of the study to obtain infective larvae and the genera of main GIN were morphologically determined using the keys of van Wyk and Mayhew (2013) [51].

### 4.5. Statistical Analysis

A factorial treatment design was formulated where the main effects were energy at two levels: high energy (*n* = 24) and low energy (*n* = 24). The other main effect was the protein with high (*n* = 24) and low (*n* = 24) levels. The effect of the physiological stage was also analyzed separating pregnant (*n* = 24) and non-pregnant ewes (*n* = 24). The EPG counts were transformed to logarithms (Log_10_ EPG + 1) and the data analyzed with a repeated measures analysis using the Mixed procedure of SAS [52], under the following statistical model.
*Y*_ijklmn_ = µ + €_i_ + Ƨ(€)_ij_ + ω_k_ + ρ_l_ + γ_m_ + ω*ρ_kl_ + ω*€_ik_ + ρ*€_il_ + ω*ρ*€_ilk_ + ω*Ƨ(€)_ijk_ + ρ*Ƨ(€)_ijl_ + ε_ijklmn_
where: *Y*_ijklmn =_ response variable (EPG, PCV, EOS, etc.); µ = general mean; €_i_ = effect of physiological stage (I = pregnancy, lactation, non-pregnant); Ƨ(€)_ij_ = fixed effect of nested week on physiological stage (j = 15, 16, 17…21 of pregnancy and 1, 2, 3…7 of lactation); ω_k_ = effect of energy level (k = high and low), ρ_l_ = effect of protein level (l = high and low); γ_m_ = random effect of ewe; ω*ρ_kl_ = interaction between energy and protein; ω*€_ik_ = interaction between energy and physiological stage; ρ*€_il_ = interaction between protein and physiological stage; ω*ρ*€_ilk_ = interaction between energy, protein and physiological stage; ω*Ƨ(€)_ijk_ = interaction of energy nested in the physiological stage; ρ*Ƨ(€)_ijl_ = interaction between protein and week nested in physiological stage; and ε_ijklmn_ = residual term from repeated measures.

## 5. Conclusions

It is concluded that a high protein level (15% crude protein) on adult productive Pelibuey ewes in tropical conditions allows the reduction of the effect of gastrointestinal nematodes during lactation. High levels of dietary protein can improve the natural resistance of ewes during peripartum rise under grazing conditions by decreasing the fecal egg count peak. In addition, productive ewes with higher dietary protein intake had an increased packed cell volume compared to productive animals fed with a lower protein diet. Physiological stage is the factor that most affects the parasitological and hematological response of Pelibuey ewes. Therefore, protein supplementation during lactation will help ewes maintain their packed cell volume levels. The non-productive ewes maintained a certain resilience and resistance to gastrointestinal parasites compared to the ewes in production, and the most critical weeks were those corresponding to the last third of gestation and the first weeks of lactation.

## Figures and Tables

**Figure 1 pathogens-11-00941-f001:**
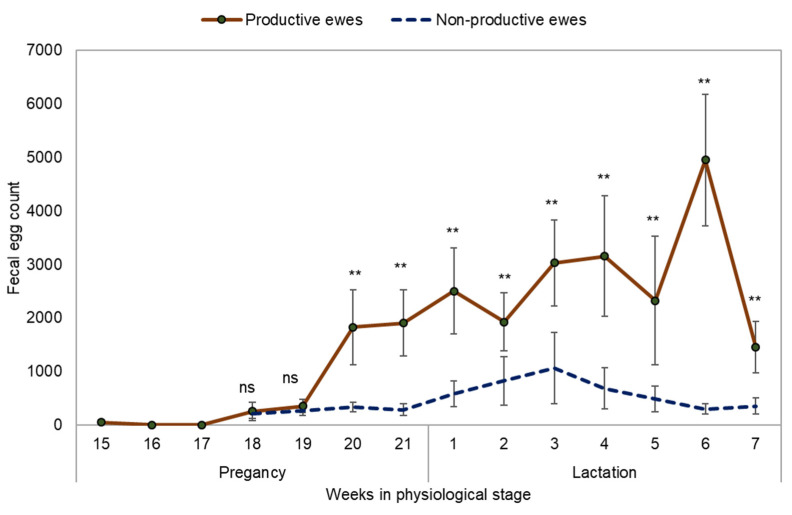
Dynamics of fecal egg count in productive and non-pregnant Pelibuey ewes infected with gastrointestinal nematodes at grazing. ns—not significant differences, **—highly significant differences (*p* < 0.01).

**Figure 2 pathogens-11-00941-f002:**
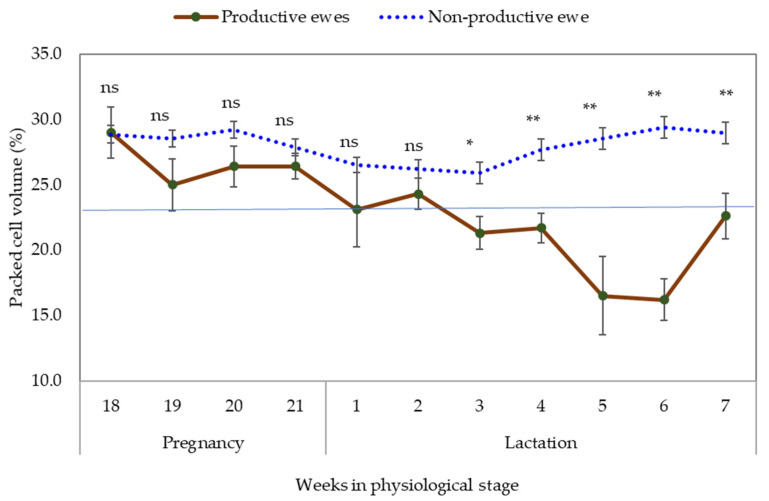
Evaluation of packed cell volume in productive and non-pregnant Pelibuey ewes infected in grazing with gastrointestinal nematodes. Blue line: threshold. ns—not significant differences, *—significant differences (*p* < 0.05), **—highly significant differences (*p* < 0.01).

**Figure 3 pathogens-11-00941-f003:**
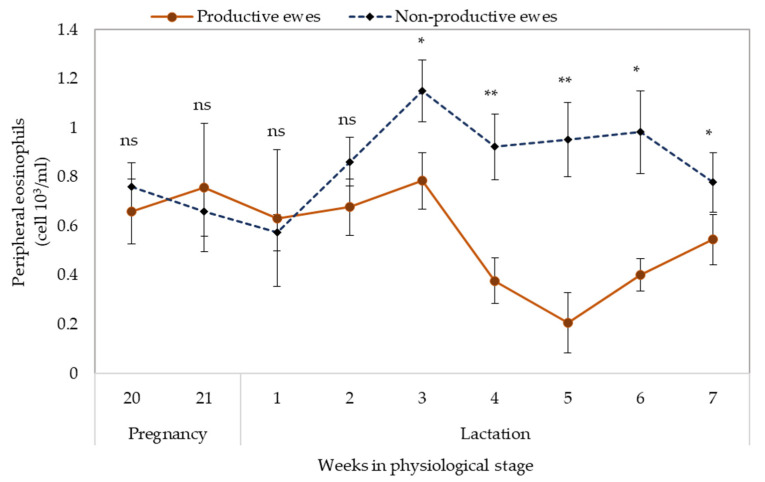
Dynamics of peripheral eosinophil count in productive and non-productive Pelibuey ewes infected in grazing with gastrointestinal nematodes. ns—not significant differences, *—significant differences (*p* < 0.05), **—highly significant differences (*p* < 0.01).

**Figure 4 pathogens-11-00941-f004:**
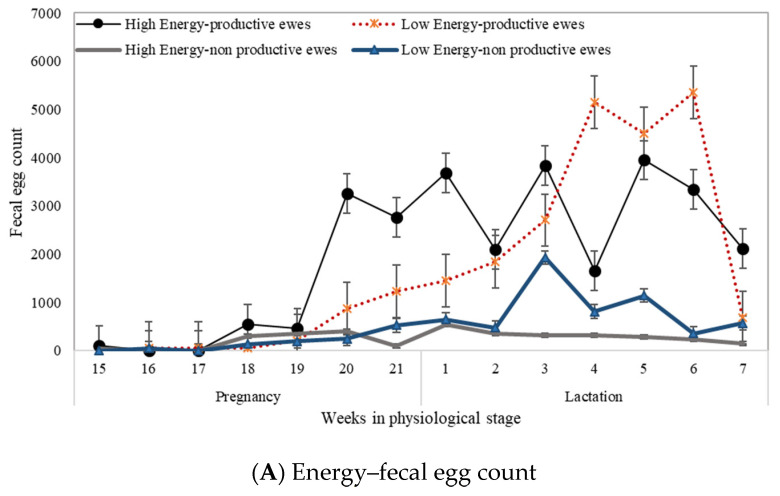
Interaction of dietary energy and protein in the physiological stage on fecal egg count and packed cell volume in productive and non-productive Pelibuey ewes under grazing conditions. (**A**) Interaction of dietary energy on fecal egg counts; (**B**) Interaction of dietary protein on fecal egg counts; (**C**) Interaction of dietary energy on packed cell volume; (**D**) Interaction of dietary protein on packed cell volume. The values are represented as mean ± standard error.

**Table 1 pathogens-11-00941-t001:** Changes in fecal nematode egg counts, packed cell volume, and eosinophil counts in productive Pelibuey ewes in comparison with non-productive ewes during peripartum.

Physiological Stage		Fecal Egg Counts		Packed Cell Volume (%)		Peripheral Eosinophils (Cells × 10^3^ µL)	
	Week	Change	Percentage ^δ^	Change	Percentage ^δ^	Change	Percentage ^δ^
Pregnancy	18	44	21.0 ^ns^	0.14	0.5 ^ns^	-	-
	19	81	29.7 ^ns^	−3.53	−12.4 ^ns^	-	-
	20	1493	447.8 **	−2.80	−9.6 ^ns^	−0.10	−13.1 ^ns^
	21	1623	572.7 **	−1.45	−5.2 ^ns^	0.10	14.7 ^ns^
Lactation	1	1915	326.0 **	−3.40	−12.8 ^ns^	0.06	10.3 ^ns^
	2	1099	132.9 **	−1.90	−7.3 ^ns^	−0.18	−21.4 ^ns^
	3	1963	184.3 **	−4.60	−17.8 *	−0.37	−31.8 *
	4	2476	363.2 **	−5.95	−21.5 **	−0.54	−59.0 **
	5	1835	374.8 **	−12.03	−42.2 **	−0.75	−78.3 **
	6	4652	1562.6 **	−13.20	−44.9 **	−0.58	−59.1 *
	7	1104	315.4 **	−6.36	−22.0 **	−0.23	−29.9 *

**^δ^** Percentage increase (positive) or decrease (negative) of the variable in productive ewes in comparison with non-productive ewes. ^ns^—not significant differences, *—significant differences (*p* < 0.05), **—highly significant differences (*p* < 0.01).

**Table 2 pathogens-11-00941-t002:** Principal effects of four different diets based on two levels of energy and protein on fecal nematode egg count, packed cell volume, and peripheral eosinophil count in productive and non-productive Pelibuey ewes infected with gastrointestinal nematodes.

Effect	Fecal Egg Counts		Packed Cell Volume (%)		Peripheral Eosinophils Cells × 10^3^ µL	
	Mean	SE	Mean	SE	Mean	SE
Protein level	*		*		^ns^	
High	837 ^a^	141	26.5 ^a^	0.46	0.73 ^a^	0.59
Low	1680 ^b^	248	24.6 ^b^	0.59	0.77 ^a^	0.74
Energy level	^ns^		^ns^		^ns^	
High	1271 ^a^	205	25.0 ^a^	0.54	0.77 ^a^	0.81
Low	1263 ^a^	209	25.9 ^a^	0.55	0.72 ^a^	0.50
Physiological stage	**		**		**	
Pregnancy	1085 ^b^	264	26.7 ^a^	0.73	0.71 ^ab^	1.48
Lactation	2709 ^c^	359	21.9 ^b^	0.69	0.59 ^b^	0.56
Non-productive	427 ^a^	96	27.7 ^a^	0.4	0.90 ^a^	0.79

^a,b,c^—Different letters in the column for each effect, differ significantly (*p* ≤ 0.05). SE—standard error. ^ns^—not significant differences, *—significant differences (*p* < 0.05), **—highly significant differences (*p* < 0.01).

**Table 3 pathogens-11-00941-t003:** Ingredients of the experimental diet used in peripartum and lactation in productive and non-productive Pelibuey ewes.

	Low Energy		High Energy	
Item	Low	High	Low	High
	protein	protein	Protein	protein
Ingredients, g/kg				
Cornmeal	30	30	260	205
Soybean paste flour	50	200	10	150
Sugar cane	830	650	670	600
Mineral mix	5	5	5	5
Urea	15	20	15	20
Molasses	0	0	40	20
Wheat bran	70	95	0	0
Chemical composition				
Dry Matter (g/kg)	349	433	501	545
Crude Protein, g/kg DM	87.4	150	85.5	154
Neutral Detergent Fiber, g/kg DM	609	540	497	479
Acid Detergent Fiber, g/kg DM	379	317	304	284
Ethereal Extract, g/kg DM	7.2	7.9	7.9	7.5
Crude Ashes, g/kg DM	52.6	54.6	44.9	47.5
Metabolizable Energy [46], MJ/kg DM	9.59	9.72	10.26	10.18

## Data Availability

Not applicable.

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
