# Peer review of "Protein Supplementation as a Nutritional Strategy to Reduce Gastrointestinal Nematodiasis in Periparturient and Lactating Pelibuey Ewes in a Tropical Environment"

_pathogens, 2022, doi:10.3390/pathogens11080941_

Round 1

Reviewer 1 Report

This is an interesting study that differentiates the effects of protein in the pregnant and lactational periods.

The main problem understanding the impact of this study is in the clarity of the design and presentation format for the results. With respect to this, there needs to be more information on when after mating the pregnancy was diagnosed and how, when the groups were selected and run together on pasture, was there any drenching applied (as would be conducted before the third trimester) and whether the supplementary food provided was fully consumed.

To assist with clarity in the results, Table 1 is not really necessary and could be replaced by a table outlining the cumulative FWEC for the 4 treatment groups including each of their 6 non-pregnant counterparts. Fig 4 has the results for combined analyses (with Stats), and for all figures, the sentence "the results are expressed as mean FWEC +/- SD or SE [not mentioned]" and " differences in superscripts within columns differ significantly P< xxx". This should also be more specifically mentioned in the text (eg line 100; "very large"")

Specific comments: were L3 cultures performed routinely or simply before the trial? If so, then did the pasture composition change during the trial as we are expecting field pick-up? If rotational grazing was done, how often (ie before the Hc pre-patent period?). I am assuming that all 48 sheep were grazed together for the 6h period? However, housing them for the remaining 18h with feed does enable those with less energy to graze, to have a welcome meal and rest. But that comes with the design, so not a big consideration.

lines 113-120 are ambiguous as energy is introduced into the section on protein.

were any worm counts preformed to check for inhibited, immature and adult stages as FWEC and L3 culture rely on adults?

The references are heavily biased to Hc, so the protein concept for relief of the PPRI is not new, but this study is confirmatory. Rather than  a benefit to immunity (which does not operate against the mobile adults which have already escaped suppression), it is likely assists with maintenance of the PCV, giving the ewe more energy to graze. however, the effect of the protein supplementation on the lactational period is a very nice result!

If one needs a more foundation study of the effects of protein supplementation during the PPRI, then try Donaldson et al (1997), Proc NZ Soc Animal Production 57: 186-9.

J, Donaldson, MFJ Van Houtert, and AR Sykes

Proceedings of the New Zealand Society of Animal Production, Volume 57, , 186-189, 1997

Author Response

Response to review

This is an interesting study that differentiates the effects of protein in the pregnant and lactational periods.

The main problem understanding the impact of this study is in the clarity of the design and presentation format for the results. With respect to this, there needs to be more information on when after mating the pregnancy was diagnosed and how, when the groups were selected and run together on pasture, was there any drenching applied (as would be conducted before the third trimester) and whether the supplementary food provided was fully consumed.

Author: The pertinent changes were made in the methodology to respond to this approach.

To assist with clarity in the results, Table 1 is not really necessary and could be replaced by a table outlining the cumulative FWEC for the 4 treatment groups including each of their 6 non-pregnant counterparts.

Author: This table was not removed at the suggestion of another reviewer and only the statistical differences were added. To respond to this aspect, four more graphs were made showing the groups by treatment and their comparison with the non-productive sheep.

Fig 4 has the results for combined analyses (with Stats), and for all figures, the sentence "the results are expressed as mean FWEC +/- SD or SE [not mentioned]" and " differences in superscripts within columns differ significantly P< xxx". This should also be more specifically mentioned in the text (eg line 100; "very large"").

Author: Corrections in tables and figures were added.

Specific comments: were L3 cultures performed routinely or simply before the trial? If so, then did the pasture composition change during the trial as we are expecting field pick-up? If rotational grazing was done, how often (ie before the Hc pre-patent period?). I am assuming that all 48 sheep were grazed together for the 6h period? However, housing them for the remaining 18h with feed does enable those with less energy to graze, to have a welcome meal and rest. But that comes with the design, so not a big consideration.

Author: L3 cultures were only done at the beginning of the study and confirmation was no longer done throughout the study period. Perhaps stool samples should have been taken during the study period, but it was not possible to do so. In other previous studies, it has been seen that Haemonchus contortus has prevailed throughout the year in a similar percentage. The ewes remained together after the breeding period, which was added in the document. Perhaps the rest period during housing is the reason why the energy level was not as important in the control of nematodes as other studies have indicated. This is taken up again in the discussion.

lines 113-120 are ambiguous as energy is introduced into the section on protein.

Author: The reference to energy was removed from the text.

were any worm counts preformed to check for inhibited, immature and adult stages as FWEC and L3 culture rely on adults?

Author: Only L3 were checked in the initial cultures. To count inhibited or immature larvae and for adults, sheep sacrifice or laparoscopy is required, techniques that were not within the scope of the study.

The references are heavily biased to Hc, so the protein concept for relief of the PPRI is not new, but this study is confirmatory. Rather than a benefit to immunity (which does not operate against the mobile adults which have already escaped suppression), it is likely assists with maintenance of the PCV, giving the ewe more energy to graze. however, the effect of the protein supplementation on the lactational period is a very nice result!

Author: Yes, it was surprising that the energy was not limiting, but I think that, as you suggest, the confinement time was long and the sheep were not allowed to graze any longer, which surely had an influence.

Reviewer 2 Report

Article: “Protein Supplementation as a Nutritional Strategy to Reduce the Gastrointestinal Nematodiasis in Periparturient and Lactaing Pelibuey Ewes”

The manuscript describes an in vivo study investigating the effect of protein and energy supplementation in Pelibuey ewes of different physiological stages on gastrointestinal (GI) nematode infections and animal resilience. The study is well conducted and the manuscript is generally well-written, and it investigates an important complementary parasite control option that can be employed to reduce the reliance on anthelmintics in small ruminants.

However, the manuscript needs major corrections and re-analysis of statistical differences before it can be accepted for publication. Moreover, at first sight the manuscript lacks novelty, as the topic investigated in this study is already well-known from previous studies: that a higher protein/energy intake enhances the resilience of sheep towards GI nematode infections. This has been confirmed in previous studies both from temperate and tropical regions, including the use of Pelibuey sheep (e.g. https://doi.org/10.1017/S1751731111001339; https://doi.org/10.1016/j.heliyon.2020.e05870)

Nevertheless, what I think is novel of the present work is the testing of the effect of protein/energy supplementation towards GI nematodes in adult (peripartum and lactating) Pelibuey ewes in tropical conditions. In that way, I believe the article gains in novelty and relevance.

Therefore, the following comments are given for the different sections of the manuscripts, for the authors to prepare a revised version of the manuscript.

Title:

- Please add a reference to the “tropical environment” or “tropical farming conditions” in the title.

Abstract:

- Line (L) 25-26: the high/low energy groups are swapped, please correct.

- L29-31: Please re-write this section after the new statistical analyses requested on each week of the study, instead of the averaged FEC/PCV data from all the study period (please see comments below).

Introduction/Aim:

- Based on my initial comment, I suggest to the authors to re-write the introduction and aim of the manuscript, to focus on testing the effects of protein and energy supplementation in productive vs non-productive Pelibuey ewes in tropical farming conditions (i.e. Equatorial savanna).

Materials and methods:

- L236: Were the ewes assigned to the different groups randomly? Were the ewes allocated in groups with similar initial FEC between groups? Or what was the criteria used to form similar groups for the study (similar weight?)? Please add this information.

- Table 3: The high energy diets also had a higher DM than the low energy diets. How this could have affected the effects observed (e.g. in terms of voluntary intake)? Please comment on this in the discussion.

- Table 3: There are several variables to describe the energy content of the experimental diets, and is difficult to judge whether the diets had indeed different energy contents. Please just select and present one variable to describe the energy content of the diets in Table 3.

- 4.5 Statistical analyses:

It is incorrect to log transform count data (i.e. FEC) before statistical analyses (please see: https://doi.org/10.1016/j.vetpar.2007.01.009; https://doi.org/10.1111/j.2041-210X.2010.00021.x).

The authors should therefore repeat the statistical analyses of the untransformed FEC data using an appropriate model, such as generalized linear mixed effect model with a binomial or Poisson distribution. However, for this analysis the authors will need to compare the FEC between animal groups at each time point (week), and not as a repeated measures for the entire study period. As the authors have used SAS, there is the “GLIMMIX procedure” for generalized linear mixed effect models (see the following article for guidance: https://doi.org/10.1016/j.tree.2008.10.008).

If the other variables (PCV, EOS, etc) are normally distributed, then is OK to analyse the data using "normal" linear mixed models (such as SAS PROC MIXED).

Furthermore, please analyse the statistical differences of FEC, PCV and EOS between productive vs non-productive ewes, to add the missing data in Table 1 and Fig 1-3 (see comments below).

Results:

- Fig. 1: With the new generalized linear mixed effect model for FEC, it would be possible for the authors to confirm whether there are significant differences at each time point (week) between animal groups. These differences should be included in the graph.

- Fig. 2: Please add the statistical differences between groups.

- Fig. 3: Please add the statistical differences between groups.

- Table 1: Please add the statistical significance of the percentage change in FEC, PCV and EOS for each week between animal groups.

- Table 2: What does the column "mean FEC/PCV/EOS" reflect? The averaged FEC/PCV/EOS of all weeks? This is highly misleading, as the authors have described the high temporal variations of these variables. Moreover, the data of both productive and non-pregnant ewes seem to have been merged, which hide the high variations between these animal groups not related with the diets effect. Therefore, please make two different tables from the original Table 2: one table for the productive ewes and a second table for the non-pregnant ewes. Moreover, the statistical differences and the diet effects in these new tables should also be presented per week during the pregnancy and lactation weeks (as presented already in Table 1).

- Fig. 4: Please modify these graphs and present the data as Fig. 1/Fig. 2, with the weeks in physiological state in the X axis and the FEC/PCV values in the Y axis. The presentation of the averaged FEC/PCV data of the whole study period is misleading, as the important temporal variations are missing. It may be necessary to make different graphs also for the different physiological states (pregnant-lactating-not-pregnant). By presenting the data per week, it would be more obvious for the reader that the diet effects are circumscribed to specific time points in productive ewes.

Discussion:

- L156: Low peripheral eosinophils only suggest a reduction in immune response, but do not confirm it. The immunity towards GI nematodes is mainly localised at the abomasal/intestinal mucosa, and a confirmation will require the investigation of immune cells in that tissues post-mortem.

- L157: Please replace “his” by “this” and “be” by “is”

- L179-181: Based on what result do the authors state that the energy levels "provided were sufficient to mount an adequate immune response"? In Table 2 the FEC, PCV and EOS  between high and low energy level groups were not statistically different. Please delete that statement, or otherwise provide an argument to keep it.

- L190-191: This statement gives the impression that the authors consider that the energy levels between high/low energy groups in their study were indeed similar. Is this the case? If this is the case, please clarify this issue and provide a critical assessment on why it was not possible to achieve different energy levels in the high/low energy diets. In this regard, please also discussed the importance of energy intake and rumen fermentable energy in the resilience of small ruminants farmed towards GI nematodes in tropical conditions (see https://doi.org/10.1016/bs.apar.2016.02.025 [section 2.4] and https://doi.org/10.1051/parasite/2015019)

- L221: Please replace "always been" by "commonly been reported to be"

Conclusions:

- Please re-write the whole conclusion section based on the new analyses and results observed and on my previous comment above on the aim of the study (effect on adult productive Pelibuey ewes in tropical conditions)

Author Response

Response in the attached

Round 2

Reviewer 2 Report

Revised version of the manuscript: “Protein Supplementation as a Nutritional Strategy to Reduce the Gastrointestinal Nematodiasis in Periparturient and Lactaing Pelibuey Ewes”

The revised manuscript has greatly improved the presentation of the work and the authors have considered most of my comments to the earlier version. I have the following minor comments that should be address before publication:

Title:

- Please add “in a Tropical Environment”

Abstract:

- L27: Please replace “two levels of protein (high, with 15%, n = 24; and low, with 8 %, n = 24)”, by “two levels of protein (high, 15% crude protein in diet, n = 24; and low, 8% crude protein in diet, n = 24)”

-L30- 32: Please replace “The high dietary protein treatment showed a reduction in FEC during lactation and improved the PCV than the low-protein level” by “The high dietary protein level had a significant effect on reducing the FEC and increased the PCV of ewes during lactation, in comparison with animals fed with the low protein level”

- L35: Delete “during pregnancy”

Introduction:

- L55-57: Please add a reference for this statement.

- L77-78: Please replace wording with “….damage to animal health. The objective of the present study was to…..”

Materials and methods:

- L289: Replace “hectarea” by “hectare”

- L318: Replace by “…according to…”

Results:

- Fig. 1, Fig. 2 and Fig. 3: Please just present the graphs with statistical differences

- Table 1:

Please change the title of the Table 1 to: “Changes in fecal nematode egg counts, packed cell volume, and eosinophil counts in productive Pelibuey ewes in comparison with non-productive ewes during the peripartum”

Please change the footnote to “*Percentage increase (positive) or decrease (negative) of the variable in productive ewes, in comparison with non-productive ewes”

- Table 2: Please specify in the title whether data from both productive and non-productive ewes are presented as one, or if it is data from just productive or non-productive ewes

 - Fig. 4: Please add the subtitle of each graph in the main title of the Figure, with a different letter for each graph (A, B, C, D), as for example: “Figure 4. Interaction of dietary energy and protein level and the physiological stage on fecal egg count and packed cell volume in Pelibuey ewes under grazing conditions. A. Interaction of dietary energy on fecal egg counts. B. Interaction of dietary protein on fecal egg counts, C.….”. In addition, please also indicate that only dietary protein had a significant effect on the variables.

- Section 2.2: Please add here how much higher was the protein level in the high vs low protein group. For example: “In the present study ewes receiving the high protein diet (15% crude protein in diet) showed lower FEC and higher PCV values (p ≤ 0.05) than ewes feeding a low protein level (8% crude protein in diet), independent of energy”.

Discussion:

- L180: Please replace “resistance of H. contortus to main chemical drugs” with “increasing drug resistant level in H. contortus populations”

- L180: Replace “specie” by “species”

- L183-184: Please replace with “as suggested in our study by the low peripheral eosinophils counts, particularly in productive and heavily infected animals”

- L226: Please replace “….energy level did not show differences in the three variables…” by “…energy level had no significant effects on the three variables….”

Conclusions:

- L353: Please replace “…that protein level…” by “…that a high protein level (15% crude protein in the diet)….”

- L356-357: Please replace “Also PCV had less reduction than ewes with a low level of protein in the diet.” by “In addition, productive ewes with a higher dietary protein intake had an increased PCV than productive animals fed with a lower protein diet”.

Author Response

Thank you very much for the review, we agree with the suggested observations. Suggested changes and a few additional small changes have been made.

Title:

- Please add “in a Tropical Environment”

Author: It had been corrected from the first time. However, in the record it appears as originally sent.

 Abstract:

 - L27: Please replace “two levels of protein (high, with 15%, n = 24; and low, with 8 %, n = 24)”, by “two levels of protein (high, 15% crude protein in diet, n = 24; and low, 8% crude protein in diet, n = 24)”

Author: Suggested changes were made to the text.

-L30- 32: Please replace “The high dietary protein treatment showed a reduction in FEC during lactation and improved the PCV than the low-protein level” by “The high dietary protein level had a significant effect on reducing the FEC and increased the PCV of ewes during lactation, in comparison with animals fed with the low protein level”

Author: Suggested changes were made to the text.

- L35: Delete “during pregnancy”

 Author: You are right about the wording error. Thank you very much. Suggested changes were made to the text.

Introduction:

 - L55-57: Please add a reference for this statement.

Author: we added a very specific reference.

- L77-78: Please replace wording with “….damage to animal health. The objective of the present study was to…..”

Author: Suggested changes were made to the text

Materials and methods:

- L289: Replace “hectarea” by “hectare”

Author: Suggested changes were made to the text

- L318: Replace by “…according to…”

Author: Suggested changes were made to the text 

Results:

- Fig. 1, Fig. 2 and Fig. 3: Please just present the graphs with statistical differences

The graphs have the statistical differences. If you could help us by being more specific

- Table 1:

Please change the title of the Table 1 to: “Changes in fecal nematode egg counts, packed cell volume, and eosinophil counts in productive Pelibuey ewes in comparison with non-productive ewes during the peripartum”

Please change the footnote to “*Percentage increase (positive) or decrease (negative) of the variable in productive ewes, in comparison with non-productive ewes”

Author: Suggested changes were made to the text

- Table 2: Please specify in the title whether data from both productive and non-productive ewes are presented as one, or if it is data from just productive or non-productive ewes

Author: Suggested changes were made to the text

 - Fig. 4: Please add the subtitle of each graph in the main title of the Figure, with a different letter for each graph (A, B, C, D), as for example: “Figure 4. Interaction of dietary energy and protein level and the physiological stage on fecal egg count and packed cell volume in Pelibuey ewes under grazing conditions. A. Interaction of dietary energy on fecal egg counts. B. Interaction of dietary protein on fecal egg counts, C.….”. In addition, please also indicate that only dietary protein had a significant effect on the variables.

Author: Suggested changes were made to the text

- Section 2.2: Please add here how much higher was the protein level in the high vs low protein group. For example: “In the present study ewes receiving the high protein diet (15% crude protein in diet) showed lower FEC and higher PCV values (p ≤ 0.05) than ewes feeding a low protein level (8% crude protein in diet), independent of energy”.

Author: Suggested changes were made to the text

Discussion:

- L180: Please replace “resistance of H. contortus to main chemical drugs” with “increasing drug resistant level in H. contortus populations”

Author: Suggested changes were made to the text

- L180: Replace “specie” by “species”

Author: Suggested changes were made to the text

- L183-184: Please replace with “as suggested in our study by the low peripheral eosinophils counts, particularly in productive and heavily infected animals”

Author: Suggested changes were made to the text

- L226: Please replace “….energy level did not show differences in the three variables…” by “…energy level had no significant effects on the three variables….”

Author: Suggested changes were made to the text

Conclusions:

- L353: Please replace “…that protein level…” by “…that a high protein level (15% crude protein in the diet)….”

Author: Suggested changes were made to the text

- L356-357: Please replace “Also PCV had less reduction than ewes with a low level of protein in the diet.” by “In addition, productive ewes with a higher dietary protein intake had an increased PCV than productive animals fed with a lower protein diet”.

Author: Suggested changes were made to the text